# Edit As You Wish: Video Caption Editing with Multi-grained User Control

## ABSTRACT

Automatically narrating videos in natural language complying with user requests, i.e. Controllable Video Captioning task, can help people manage massive videos with desired intentions. However, existing works suffer from two shortcomings: 1) the control signal is single-grained which can not satisfy diverse user intentions; 2) the video description is generated in a single round which can not be further edited to meet dynamic needs. In this paper, we propose a novel **V**ideo **C**aption **E**diting **(VCE)** task to automatically revise an existing video description guided by multi-grained user requests. Inspired by human writing-revision habits, we design the user command as a pivotal triplet {*operation, position, attribute*} to cover diverse user needs from coarse-grained to fine-grained. To facilitate the VCE task, we *automatically* construct an open-domain benchmark dataset named VATEX-EDIT and *manually* collect an e-commerce dataset called EMMAD-EDIT. We further propose a specialized small-scale model (i.e., OPA) compared with two generalist Large Multi-modal Models to perform an exhaustive analysis of the novel task. For evaluation, we adopt comprehensive metrics considering caption fluency, command-caption consistency, and video-caption alignment. Experiments reveal the task challenges of fine-grained multi-modal semantics understanding and processing. Our datasets, codes, and evaluation tools are ready to be open-sourced.

## CCS CONCEPTS

• **Computing methodologies** → **Natural language generation**.

## KEYWORDS

Video Captioning, Caption Editing, Controllable Generation

## 1 INTRODUCTION

The proliferation of videos on the Internet heralds the era of video-dominated media. Video captioning, i.e. automatically describing videos using natural language, has been a prevalent task to assist people in comprehending and managing massive videos. However, conventional video captioning systems [46, 58] tend to generate intention-agnostic descriptions, ignoring the various demands of different users. Therefore, a new task branch, namely controllable video captioning [5, 8, 22, 60], has been proposed to integrate user intention as a control signal to guide the description generation.

**Unpublished working draft. Not for distribution.**

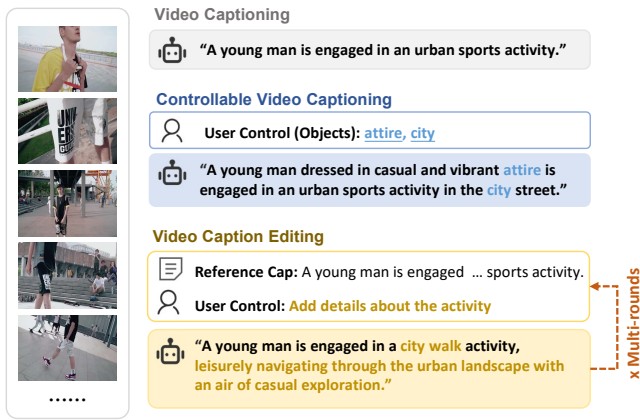

**Figure 1: Comparisons between our proposed Video Caption Editing (VCE) task with conventional video captioning and controllable video captioning.**

Although controllable video captioning has great potential in practical applications, existing works have two non-negligible drawbacks. First, they all employ fixed control signals that can only express *single-grained controls*, such as Part-of-Speech(POS) [48] for structure control, or specified object tags [22] for semantic control. These single-grained controls can not satisfy flexible and diverse user demands. Second, they are *single-round controls* that generate a video description once which can not be further revised. Whereas iteratively revising sentences until the ideal texts is a natural process for humans [10]. Imagine a real-world scenario, such as E-commerce product promotion, where sellers upload product videos with descriptions to attract customers. There is a good chance that automated video descriptions fail to highlight the seller's preferences. As a result, sellers need to further improve the descriptions by themselves, which is time-consuming and labor-intensive, especially when facing massive long videos.

We propose a novel **Video Caption Editing (VCE)** task to automate the video description editing process. The task aims to edit an existing video description conditioned on user commands and video content. As depicted in Figure 1, the inputs of VCE task consist of a video, a reference description, and a user control. It outputs an edited video description based on the user command as a control signal. The reference caption can be initialized using the last edited output sentence which can thus enable *multi-round modification*. The VCE task can facilitate personalized video description generation by fulfilling miscellaneous demands from different users or dynamic demands from the same user.

In the VCE task, how to define the user command to cover multi-grained requests is crucial. Inspired by the human writing-revision habits, we unify the user edit commands into a triplet format {*operation, position, attribute*} (depicted in Figure 2) for three advantages.

Firstly, it condenses the core elements in an editing operation. Secondly, it can accommodate two prevalent front-end interface signals including natural language and editing trajectories from tablet computers. Finally, the different combinations of three elements in the triplet can cover *multi-grained* user commands from coarse-grained control (e.g. sentence length change) to fine-grained control (e.g. insert new details in the specific position), as illustrated in Table 1.

We collect two novel benchmark datasets named VATEX-EDIT and EMMAD-EDIT to support the exploration of the VCE task. The VATEX-EDIT dataset is automatically constructed from a large-scale video-text dataset VATEX [49] in the open domain. Meanwhile, to close the gap between research advances and real-life applications, we manually collect an e-commerce editing dataset called EMMAD-EDIT, which is more challenging from two aspects: 1) longer videos (average 27.1 seconds) and longer captions (average ~100 words); 2) external domain knowledge needed to generate product-oriented video descriptions. Based on the two benchmark datasets, we propose a specialized model, namely OPA, that converts the command triplet into a textual token sequence to alleviate heterogeneity among multi-grained commands. We demonstrate the feasibility of utilizing a unified framework to handle seven types of user commands. Moreover, we adopt two generalist Large Multimodal Models (LMMs) as a comparison to gain an in-depth understanding of the characteristics and challenges of the VCE task.

The main contributions of this paper are four-fold. 1) To the best of our knowledge, we are the first to propose the VCE task to achieve *multi-round* editing and design the user command as a triplet format to express *multi-grained* user requests. 2) We build two benchmark datasets from different domains, including the general domain (VATEX-EDIT) and commercial domain (EMMAD-EDIT), to facilitate the investigation of the VCE task. 3) We develop an evaluation suite to assess the edited video description based on caption fluency, command-caption consistency, and video-caption alignment. 4) We propose a unified specialist framework OPA and adapt two generalist LMM methods to initially tackle the task, followed by a comprehensive analysis.

## 2 RELATED WORK

**Controllable Video Captioning.** Video captioning [1, 25, 30, 33, 37, 39, 43, 46, 58] is a challenging cross-modal task to automatically describe the visual contents of a video in natural languages. In order to satisfy the varied pragmatic interests of different users, controllable video captioning [5, 8, 22, 60] has been a newly prevalent task. It aims to derive video descriptions conditioned on a predefined control signal, e.g. visual object tags. Wang et al. [48] introduce Part-of-Speech(POS) information as guidance and Yuan et al. [60] directly utilize an exemplar sentence. Their goal is to generate descriptions with desired syntactic structures. Meanwhile, other works aim to control sentence semantics. Chen et al. [5] proposes a topic-guided model to generate topic-oriented descriptions. Liu et al. [22] focus on producing object-oriented sentences controlled by multiple user-interested objects. However, the above endeavors all generate a sentence once and can't be edited dynamically. Besides, their designed control signals are single-grained which can not cover flexible user intentions. Instead, we define a novel VCE task that can revise a description in multiple rounds covering multi-grained user demands.

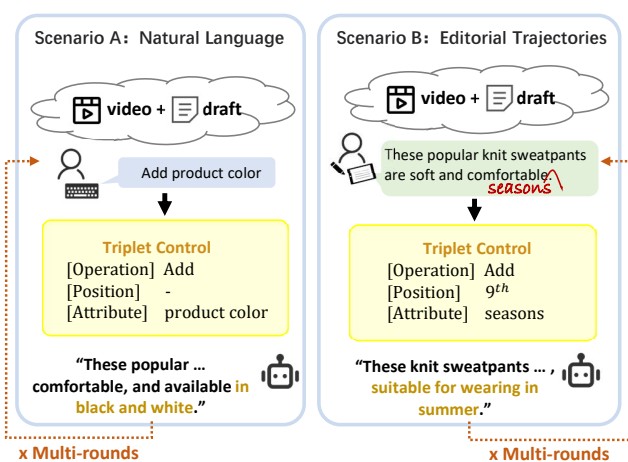

**Figure 2: The triplet control designed in the VCE task can pivot two prevalent interaction signals including natural language (Scenario A) and editing trajectories (Scenario B).**

**Image Caption Editing.** Conventional image captioning [4, 17, 20, 21, 41, 47, 52, 55, 56] generates the description for images from scratch which may lead to factual mistakes. Sammani and Elsayed [35] firstly define the image caption editing task that modifies an existing caption (a.k.a. reference caption) conditioned on the image content to obtain more accurate descriptions. Sammani and Melas-Kyriazi [36] further propose a novel EditNet framework to achieve interactive and adaptive edits. Yuan et al. [59] design an adaptive text-denoising network to alleviate the semantic gap between input images and reference sentences. The above works all edit image captions implicitly. Wang et al. [50] propose the explicit image caption editing task to make the modification process more explainable and efficient. In summary, these caption editing works can only correct the wrong content in the reference captions and ignore specific edit intentions of different users. In this paper, we integrate multi-grained user commands into the video description editing process.

**Large Multimodal Models.** Recent months have witnessed the tremendous success of Large Language Models (LLMs) [6, 27–29, 44, 45, 61] towards artificial general intelligence. Further, Large Multimodal Models (LMMs) [2, 3, 7, 15, 18, 23, 26, 34, 57, 62] endow LLMs with the visual understanding ability by incorporating vision backbones [16, 32, 42]. Existing LMMs primarily bifurcate into two categories: Image Large Language Models (ImgLLMs) [2, 3, 7, 23, 57] and Video Large Language Models (VidLLMs) [15, 18, 26, 34, 62]. The standard architecture of an LMM comprises a vision backbone for encoding images or videos, a projector [3, 13] to condense and translate visual embeddings into textual semantic space, and an LLM to process all multimodal contexts. As the VCE task involves capturing video semantics, understanding textual user controls, and enabling multi-grained text editing, it can serve as a new touchstone for LMMs.

**Table 1: Editing commands via different combination of elements *{operation, position, attribute}*. It covers seven multi-grained demands from coarse-grained controls (e.g. expand description) to fine-grained controls (e.g. add specified *attributes* at specified *positions*). The atomic operations consist of *add* and *delete*. The multi-grained commands with a reference caption (e.g. "A group of girls is on the field playing a game.") are unified as a control token sequence (Section 5.1) to guide the model.**

| Command | | | Notation | Demand | Unified Input Control |
|---|---|---|---|---|---|
| opera. | pos | attr | | | |
| ✓ | | | $\langle add, -, - \rangle$ | expand description | [ADD] A group of girls is playing a game. |
| ✓ | ✓ | | $\langle add, pos, - \rangle$ | expand description at specified *positions* | [ADD] A group of girls is [MASK] playing a game. |
| ✓ | | ✓ | $\langle add, -, attr \rangle$ | add specified *attributes* in description | [ADD] field, hockey; A group of girls is playing a game. |
| ✓ | ✓ | ✓ | $\langle add, pos, attr \rangle$ | add specified *attributes* at specified *positions* | [ADD] field, hockey; A group of girls is [MASK] playing a game. |
| ✓ | | | $\langle del, -, - \rangle$ | shorten description | [DEL] A group of girls is on the field playing a game. |
| ✓ | ✓ | | $\langle del, pos, - \rangle$ | shorten description at specified *positions* | [DEL] A group of girls is on (the filed) [MASK] playing a game. |
| ✓ | | ✓ | $\langle del, -, attr \rangle$ | delete specified *attributes* from description | [DEL] field, group; A group of girls is on the field playing a game. |

## 3 VIDEO CAPTION EDITING TASK

### 3.1 Task Definition

Given a video $V$ and a reference caption $R = \{r_1, \ldots, r_L\}$, the VCE task aims to generate an edited caption $Y = \{y_1, \ldots, y_T\}$ according to the user edit command $C$. The edited caption $Y$ should satisfy the constraints of $V$, $R$ and $C$. Given a ground truth caption $Y^*$, the maximum likelihood estimation (MLE) training objective of VCE task can be formulated as:

$$\mathcal{L}_{\text{MLE}} = -\frac{1}{T} \sum_{t=1}^{T} \log p\left(y_t^* \mid y_{<t}^*, V, R, C\right) \quad (1)$$

The reference caption can be initialized with the output caption from the last round of editing. It is also possible to start the editing process using a auto-generated sentence or human-written one as the reference. Due to the reference caption input setting, the VCE task can naturally achieve an interactive editing process with successive editing rounds, which is in line with human writing habits [10]. The interactive and multi-round revisions can help produce descriptions with higher user satisfaction.

### 3.2 User Edit Command

It is not trivial to define flexible edit commands in the VCE task to meet various realistic user needs. We observe that natural language and writing-revision traces are two natural interactive modes. The former can be received from keyboards or speech converters, while the latter conveniently expresses user intentions with the prevalence of tablets and wireless stylus pens[1]. A command representation compatible with the above two signals is important and meaningful. In this paper, we propose a novel command representation in a triplet format **{*operation, position, attribute*}**, where **operations** control the overall description editing, **positions** specify the editing locations, which could affect the syntax of sentences, and **attributes** guide the editing operation to control the semantic contents of descriptions.

We define the atomic edit operations as *add* and *delete*, considering that the *replace* editing can be decomposed into the two atomic operations (i.e. first *delete* then *add*). Meanwhile, *position* and *attribute* in the triplet are optional, therefore, as shown in Table 1, seven specific commands[2] via different combinations of *operation, position, and attribute* elements in the triplet can cover multi-grained realistic demands from coarse-grained (global) controls to fine-grained (local) controls. The designed triplet command can be flexibly obtained by processing the inputs from front-end interfaces including natural language and writing-revision traces (details in Appendix E). In the following method and experiments sections, we perform video description editing directly based on the triplet command.

## 4 DATA COLLECTION

To faciliate the novel VCE task, we *automatically* construct an open-domain dataset VATEX-EDIT, and *manually* annotate an E-commerce dataset EMMAD-EDIT. Table 2 displays the overall data statistics. Compared to prior image caption editing datasets such as COCO-EE and Flickr30K-EE, our new datasets present several distinct advantages: 1) more challenging with the video input and lengthier captions; 2) more diverse encompassing open-domain and e-commerce data; and 3) larger in scale. Specific annotated data instances are illustrated in Figure 3.

### 4.1 VATEX-EDIT Construction

It is challenging to construct data samples for the VCE task from scratch, which needs a quadruple *(video, command, reference caption, edited caption)* data, abbreviated as $(V, C, R, Y)$. To mitigate the difficulty, we build the VATEX-EDIT dataset by expanding the widely-used video captioning dataset VATEX [49], which has high-quality caption annotations for each video.

We sample an annotated caption of a video as the reference caption, and the next goal is to construct the *command* and the

---

[1]https://www.apple.com/apple-pencil/

[2]Note that we omit the command "$\langle del, pos, attr \rangle$, *delete attributes at specified positions*", as it can be covered by "$\langle del, pos, - \rangle$, *delete description at specified positions*".

**Table 2: Data statistics of VATEX-EDIT and EMMAD-EDIT dataset. # denotes the number. *VTime* refers to the average duration of videos in seconds. $Len_{Ref}$ denotes the average length of reference captions and $Len_{GT}$ is the average length of ground-truth captions. *Edit Dist* means the average minimum edit distance between reference captions and edited captions.**

| Dataset | Vision | #Videos/Images | | | #Editing instances | | | VTime | $Len_{Ref}$ | $Len_{GT}$ | Edit Dist | Vocab |
|---|---|---|---|---|---|---|---|---|---|---|---|---|
| | | Train | Val | Test | Train | Val | Test | | | | | |
| COCO-EE [50] | Image | 52,587 | 3,055 | 2,948 | 97,567 | 5,628 | 5,366 | - | 10.3 | 9.7 | 10.9 | 11,802 |
| Flickr30K-EE [50] | Image | 29,783 | 1,000 | 1,000 | 108,238 | 4,898 | 4,910 | - | 7.3 | 6.2 | 8.8 | 19,124 |
| VATEX-EDIT | Video | 25,467 | 2,935 | 5,867 | **784,805** | **91,513** | **181,638** | 10.0 | 14.4 | 16.0 | 11.9 | 21,634 |
| EMMAD-EDIT | Video | 16,176 | 5,418 | 5,502 | 47,569 | 15,914 | 16,169 | **27.1** | **91.3** | **93.7** | **17.8** | **44,725** |

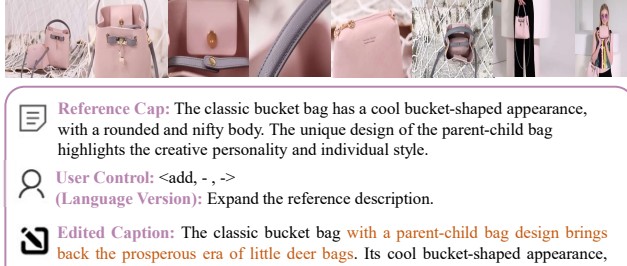

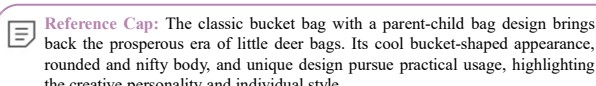

Reference Cap: The classic bucket bag has a cool bucket-shaped appearance, with a rounded and nifty body. The unique design of the parent-child bag highlights the creative personality and individual style.

User Control: <add, - , ->
(Language Version): Expand the reference description.

Edited Caption: The classic bucket bag with a parent-child bag design brings back the prosperous era of little deer bags. Its cool bucket-shaped appearance, rounded and nifty body, and unique design pursue minimalist and practical fashion, highlighting the creative personality and individual style.

Reference Cap: The classic bucket bag with a parent-child bag design brings back the prosperous era of little deer bags. Its cool bucket-shaped appearance, rounded and nifty body, and unique design pursue practical usage, highlighting the creative personality and individual style.

User Control: <add, - , 'style' >
(Language Version): Add contents about 'style' to extend the reference description.

Edited Caption: The classic bucket bag with a parent-child bag design brings back the prosperous era of little deer bags. Its cool bucket-shaped appearance, rounded and nifty body, and unique design pursue minimalist and practical fashion, highlighting the creative personality and individual style.

**Figure 3: Annotated data instances of the VCE task.**

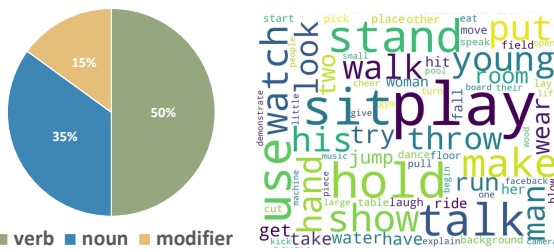

**Figure 4: Attribute statistics on the VATEX-EDIT.**

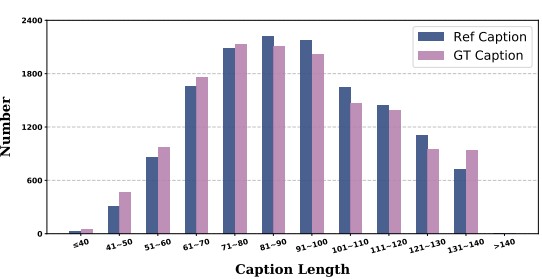

**Figure 5: Caption length distributions on EMMAD-EDIT.**

related *edited caption*. In general, we aim to construct related *(command, edited caption)* samples including: **1) coarse-grained length-control commands** referring to the global *add* or *delete* edits that result in length changes, and **2) finer-grained attribute-related commands** to achieve *add* or *delete* attributes on the reference caption.

**Coarse-grained length-control commands.** For *add* operation, we directly select a longer caption with significant length differences from the rest annotations as the edited caption. The *delete* operation does the exact opposite. Considering the *delete* operation can be easily achieved without video content referring, we replace the reference caption with negative captions that have partially misaligned semantics with the video content. Such updated quadruples require models to prioritize removing visually irrelevant content from the reference, which makes the *delete* more challenging.

**Fine-grained attribute-related commands.** We construct attribute samples in a "degradation" manner, that is firstly detecting attribute words in the reference caption $R$ and then removing them

to obtain the edited caption $Y$. We utilize Spacy Syntactic Dependency[3] and Semantic Role Labeling [38] to achieve noun, verb or modifier attributes detection and removal in $R$ while maintaining the fluency of $R_{\setminus attr}$ to get $Y$. After degradation, we can obtain quadruples for the $\langle del, -, attr \rangle$ command. We exchange the reference caption and the edited caption to obtain the opposite *add* command. The position information can be naturally recorded to support position-related commands.

**Statistics and analysis.** As shown in Table 2, compared with existing image editing datasets [50], VATEX-EDIT has two salient features: *large-scale* and *diverse*. Figure 4 visualizes the percentages of modifiers, nouns and verbs in the *attribute*. Our automatic construction strategy selects verbs as the dominant attributes because verbs are usually related to temporal visual semantics, which is also one of the core challenges of the video description task. The word cloud of specific attribute words shows the *attribute* diversity.

## 4.2 EMMAD-EDIT Collection

To satisfy realistic user-demand scenarios, we manually collect a high-quality dataset called EMMAD-EDIT in the E-commerce

---

[3]https://spacy.io/

domain based on a Chinese E-commerce video captioning dataset E-MMAD [63]. The E-MMAD dataset consists of product videos with advertising video descriptions, and additional structure information. Given a product-oriented video $V$ and an original video description $R$, we recruit crowd workers to annotate three types of edited descriptions as follows.

**Simplify original captions** to the target length while maintaining sentence fluency and coherence according to the video content. To ensure the challenge of the VCE task, we require that the length of original sentences should be reduced by at least 20%.

**Delete specific attributes.** It aims to select multiple significant attribute words/phrases from $R$ and remove the attribute-related content to get a new caption $Y$. The attributes can be nouns, verbs, or modifiers. To ensure the semantic coherence of $Y$, workers are allowed to modify other parts of $R$ following the "minimal editing" principle.

**Delete abstract attributes.** We further consider deleting abstract attributes that do not directly appear in $R$. For example, deleting "Time and Seasons" needs to locate season-related content such as "spring" and "summer". It is more challenging to edit with abstract attributes and also more down-to-earth since user intentions may be vague.

**Statistics and analysis.** To ensure annotation quality, extra workers further check the annotated cases. Table 2 shows the specific data statistics. EMMAD-EDIT has two remarkable characteristics, i.e. *long videos* and *long descriptions*. The average video length is 27.1 seconds and the average description length (specified in Figure 5) is around 100 words. We believe the challenging EMMAD-EDIT dataset will promote new technologies for the VCE task.

## 5 METHODOLOGY

In this section, we begin by introducing how to transform the triplet control into a unified textual sequence. Subsequently, we explore three approaches for the VCE task to facilitate a comprehensive comparison. We propose the **O**peration-**P**osition-**A**ttribute (**OPA**) model as a small-scale specialist. Additionally, we utilize an Image Large Language Model (ImgLLM) pipeline, and an end-to-end Video Large Language Model (VidLLM) to observe the performance of large multimodal models. Lastly, we develop an evaluation protocol for the novel task.

### 5.1 Input Format Design

We first integrate the seven specific edit commands introduced in Table 1 into a unified format to achieve multi-grained control.

The main challenge is the heterogeneity among the three elements of command, including *operation*, *position*, and *attribute*. On the one hand, *operations* and *attributes* change the textual semantics while *positions* mainly influence sentence syntax. On the other hand, *attributes* are specific textual words while *positions* are absolute position indexes.

To tackle the above challenges, we unify the input format as a textual token sequence. As shown in Table 1, we define two special tokens, [ADD] and [DEL], to represent different *add* or *delete* operations. *Attribute* words are naturally text tokens. For *position*, we put special tokens [MASK] in the reference caption to indicate the absolute position indexes. For example, a positioned reference caption "*A group of girls is* [MASK] *playing a game*" guides the model to generate new details between words "is" and "playing". Finally, we concatenate the operation token, the attribute words, and a positioned reference caption as a control sequence to guide the model for description generation. Table 1 visualizes the input control sequences under seven specific commands when the reference caption is *"A group of girls is playing a game"*.

### 5.2 OPA: A Small-Scale Model as the Specialist

We construct a small-scale encoder-decoder Transformer architecture, i.e. multi-modal BART [12], to achieve the video description editing task under the guidance of processed control sequences. We utilize the pre-trained BART weights and endow it with the multi-modal ability to understand video content. The specific architecture is depicted in Appendix A.

**Input Representation.** Given a video $V$, a reference caption $R = \{r_1, \ldots, r_L\}$, and a triplet command $C$, we first extract frame-level visual features and map them to the same dimension as textual embedding. We denote the input attributes as $A = \{a_1, \ldots, a_M\}$ and the indicated position index as $l \in [1, L]$. Taking the most fine-grained command "*add* specified *attributes* at specified *positions*" as an example, the concatenated control sequence $\widetilde{C}$ for the command is defined as $\{[ADD], A, \widetilde{R}\}$. Using special tokens to separate each part, it is formulated as:

$$\widetilde{C} = \{[\text{opera}]\,[\text{ADD}]\,[/\text{opera}]\,[\text{attr}]\,A\,[/\text{attr}]\,[\text{ref}]\,\widetilde{R}\,[/\text{ref}]\} \quad (2)$$

where the positioned reference caption $\widetilde{R}$ is formulated as:

$$\widetilde{R} = \{r_1, \ldots, r_{l-1}, [\text{MASK}], r_{l+1}, \ldots, r_L\} \quad (3)$$

Finally, we input the visual features $\{V_1, \ldots, V_N\}$ and the textual control sequence embedding $W_{\widetilde{C}} = \{W_{[\text{o}]}, W_{[\text{ADD}]}, \ldots, W_{[/\text{r}]}\}$ to the Transformer encoder. If the *position* is empty in the command, we input the original reference caption $R$. When the *attribute* is empty in the command, we set $A$ as an empty set.

**Leverage Pre-trained Knowledge.** The overall training objective as formulated in Section 3.1 is to generate an edited description conditioned on the video features and the control sequence. It is worth noting that we keep the [MASK] token consistent with the same token in the *Text Infilling* pre-trained task of BART to leverage the intrinsic pre-training textual ability.

### 5.3 LMMs as Contrastive Generalists

Large multimodal models integrate the advantages of visual understanding and remarkable natural language processing abilities (e.g., text editing) from LLMs. It is significant to probe their performance on the VCE task. Consequently, we explore two typical branches of LMMs including an ImgLLM pipeline and an end-to-end VidLLM.

**ImgLLM Pipeline.** We utilize the InstructBLIP [7] as the ImgLLM. Nevertheless, ImgLLM can only handle images or a single video frame as visual input. To adapt the ImgLLM to the VCE task, we combine InstructBLIP with ChatGPT [27]. In this way, InstructBLIP transforms visual semantics at the frame level into textual context, while ChatGPT consolidates all textual task context and achieves caption editing. Specifically, we extract frames from a given video and utilize InstructBLIP to produce detailed visual descriptions for each frame. The frame descriptions with the VCE task definition, task guidelines, and in-context demonstrations [9] of the relative

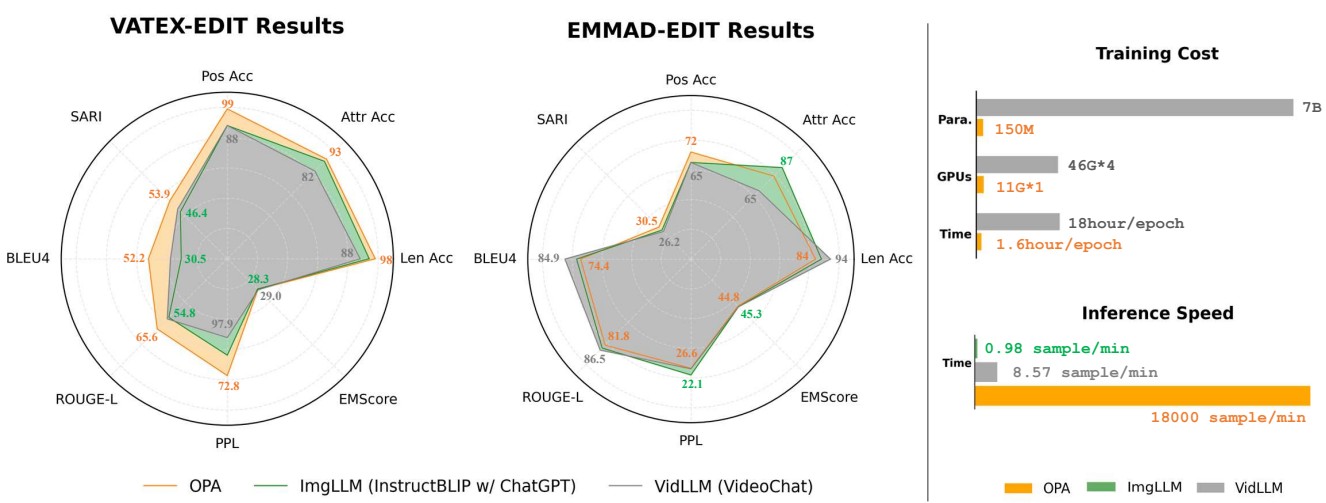

Figure 6: Overall performance of the small-scale specialist model (i.e., OPA) and large-scale generalist models (i.e., ImgLLM pipeline and end-to-end VidLLM) on the VATEX-EDIT and EMMAD-EDIT dataset. We utilize InstructBLIP [7] w/ ChatGPT [27] as the ImgLLM pipeline without training. Meanwhile, we conduct instruction tuning on the VideoChat-7B [15] as the end-to-end VidLLM method. The training cost (parameters, used GPUs, and training time) and inference speed via a single GPU are on the right. It is noted that we take the negative of PPL↓ and rescale it on the radar coordinate for better visualization.

command type will be combined as the final prompt to the ChatGPT. Instruction details are provided in Appendix A.

**End-to-end VidLLM.** As VidLLM can handle video tasks directly, we employ the VideoChat [15] as an end-to-end LMM solution. Specifically, we reformat the VATEX-EDIT and EMMAD-EDIT datasets into question-answer chat samples (refer to Appendix A for specifics) and conduct further instruct-tuning on VideoChat-7B using two datasets respectively.

### 5.4 Evaluation Suite

How to evaluate the novel VCE task is another noteworthy challenge. Conventional video captioning tasks adopt widely-used captioning metrics such as BLEU4, METEOR, and CIDEr. However, these reference-based metrics only measure the consistency between generated captions and ground-truth annotations, which are insufficient. In this paper, we evaluate the VCE task from three aspects: 1) fluency, 2) controllability, and 3) text-video alignment.

*Fluency.* Following the previous work [50], we adopt widely-used **BLEU4** [31] and **ROUGE-L** [19] metrics to measure the overall generation quality. We also use the **Perplexity (PPL)** [11] metric that reflects the grammatical correctness and semantic meaningfulness.

*Controllability.* Measuring whether an edited caption strictly follows control signals is important for the VCE task. Inspired by previous work [10], we first utilize the **SARI** [53] metric to measure the overall edit quality, i.e. the consistency between expected-to-edit and actually-edited spans. Moreover, we design three breakdown metrics namely **Length Accuracy, Attribute Accuracy** and **Position Accuracy** to measure whether the edited caption satisfies the {*operation, position, attribute*} triplet control. Concretely, Len-Acc reflects the length change accuracy. Attr-Acc checks the appearance of commanded attribute words. Pos-Acc evaluates whether the model inserts/removes content in the specified positions.

*Text-Video alignment.* The VCE task inherently requires the alignment between edited descriptions and the given video. We use **EMScore** [40] to calculate the semantic similarity between edited captions and videos. It focuses on both coarse-grained similarity (video-sentence) and fine-grained similarity (frame-word).

## 6 EXPERIMENTS

### 6.1 Implementation Details

We implement the small-scale OPA model based on Huggingface Transformers library [51]. The default setting is initialized by the $BART_{base}$. We get video frames using fps=1. For the VATEX-EDIT dataset in *English*, we adopt BLIP [14] ViT-B/16 to extract frame features. The max frame sequence N is set to 20. For the EMMAD-EDIT dataset in *Chinese*, we initialize our model with the Chinese BART. We adopt CN-CLIP [54] ViT-B-16 to extract video frame-level features. The max frame sequence N is set to 30 and the max decoding length is set to 150. For training, we use AdamW [24] with a learning rate of 1e-5 and optimize for 20 epochs with a batch size of 20. During inference, we set beam size as 5. The ImgLLM pipeline utilizes the identical frame number N as the OPA model. This pipeline doesn't involve any training. We choose one in-context learning sample for every command type integrated into the ChatGPT prompt. For VideoChat model, we set frame number N as 8 to fit its default setting. We conduct further instruct-tuning on the official 7B checkpoints with batch size 64.

### 6.2 Compare Specialist and Generalist Models

Figure 6 shows the overall performance of three baselines on the VATEX-EDIT and EMMAD-EDIT datasets. Interestingly, overall performances are divergent across the two datasets. On the large-scale open-domain VATEX-EDIT dataset, the small-scale specialist OPA model with only 150M parameters outperforms the LMM approaches. It suggests that with sufficient training instances (784,805

**Table 3: Ablation study of the OPA model on the VATEX-EDIT dataset.** *Multimodal BART* **is the backbone of OPA framework.** *Pure Transformer* **is the same model without pre-trained BART parameters.** *Vision Align* **means the vision-text alignment.**

| | Model | Controllability | | | | Fluency | | | Vision Align |
|---|---|---|---|---|---|---|---|---|---|
| | | Len-Acc | Attr-Acc | Pos-Acc | SARI | BLEU4 | ROUGE-L | PPL↓ | EMScore |
| 1 | Multimodal BART | - | - | - | 49.7 | 48.0 | 62.0 | 73.9 | 28.7 |
| 2 | Multimodal BART$_{Opera.}$ | 97 | - | - | 52.1 | 49.6 | 63.2 | 70.6 | 28.7 |
| 3 | Multimodal BART$_{Opera.+Attr}$ | 97 | 93 | - | 53.8 | 52.3 | 65.7 | 72.7 | 28.7 |
| 4 | OPA | 98 | 93 | 99 | 53.9 | 52.2 | 65.6 | 72.8 | 28.7 |
| 5 | Pure Transformer$_{Opera.+Pos+Attr}$ | 98 | 82 | 97 | 52.6 | 50.5 | 64.4 | 77.2 | 28.7 |
| 6 | 3 Single-grained Models | 96 | 69 | 99 | 53.3 | 51.5 | 64.6 | 72.8 | 28.6 |

**Table 4: Overall and breakdown performances on the EMMAD-EDIT dataset.**

| | Command | Controllability | | | | Fluency | | | Vision Align |
|---|---|---|---|---|---|---|---|---|---|
| | | Len-Acc | Attr-Acc | Pos-Acc | SARI | BLEU4 | ROUGE-L | PPL↓ | EMScore |
| 1 | ⟨ add, - , - ⟩ | 57 | - | - | 26.2 | 62.1 | 73.9 | 23.7 | 44.7 |
| 2 | ⟨ add, pos, - ⟩ | 75 | - | 59 | 27.0 | 83.6 | 90.3 | 25.1 | 44.9 |
| 3 | ⟨ add, - , attr ⟩ | 80 | 74 | - | 31.7 | 84.7 | 89.9 | 26.6 | 45.2 |
| 4 | ⟨ add, pos, attr ⟩ | 92 | 70 | 85 | 32.9 | 88.1 | 93.3 | 25.5 | 44.8 |
| 5 | ⟨ del, - , - ⟩ | 100 | - | - | 33.5 | 66.8 | 73.8 | 30.5 | 44.6 |
| 6 | ⟨ del, pos, - ⟩ | 99 | - | - | 30.7 | 83.9 | 90.6 | 28.8 | 44.6 |
| 7 | ⟨ del, - , attr ⟩ | 100 | 93 | - | 33.6 | 75.2 | 83.6 | 28.9 | 44.7 |
| 8 | Overall | 84 | 79 | 72 | 30.5 | 74.4 | 81.8 | 26.6 | 44.8 |

samples in VATEX-EDIT), a small specialized model has the potential to perform more effectively and efficiently.

On the e-commerce EMMAD-EDIT dataset, the LMM methods achieve higher scores across most metrics. EMMAD-EDIT is more challenging because it requires domain knowledge, such as unseen product attributes and advertising description style, and has limited training data (refer to Table 2). Results show that ImgLLM with InstructBLIP and ChatGPT achieve highest Attr-Acc (87%) even without training. We argue that on this domain, generalist methods are more promising to leverage their intrinsic knowledge to edit product-related descriptions.

Despite performance advantages, the training cost and inference speed must be taken into account due to the booming number of videos. As illustrated in Figure 6 (right), compared to the VidLLM and ImgLLM, the OPA model demonstrates significant benefits over both VidLLM and ImgLLM in terms of lower training costs and faster inference speeds. Designing a model that balances performance with speed and cost represents a crucial trade-off.

Although both specialist and generalist models offer unique advantages, there remains considerable scope for further enhancements to develop an effective editing system, especially on the EMMAD-EDIT dataset. The *Controllability* (Pos-Acc 72%, Attr-Acc 87% and Len-Acc 94%) on the EMMAD-EDIT dataset is insufficient. Moreover, the alignment between video and caption, as indicated by the EMScore, requires significant improvement. In conclusion, the key of the VCE task lies in the combination of fine-grained video and user control understanding and precise text editing capabilities.

## 6.3 Further Task Analysis

We conduct further ablation studies on the small-scale OPA model to delve into a detailed analysis of the characteristics of the VCE task.

**Increase control signals.** We analyze the editing performance under different control signals in Table 3. The proposed OPA framework achieves high *controllability* accuracy (Len-Acc 98%, Attr-Acc 93%, and Pos-Acc 99%) while maintaining sentence quality. The first block (lines 1-4) shows the controllability accuracy and caption quality when progressively inputting more control signals into the model. With the increasing aspects of control signals, there is no decline in sentence fluency and text-vision alignment. It indicates that our model can edit the reference caption with reasonable syntactic and semantic changes under multi-aspect guidance. Line 5 shows the result of Pure Transformer trained from scratch. Without BART pre-trained parameters, the overall controllability and fluency metrics decrease (SARI from 53.9 to 52.6, BLEU4 from 52.2 to 50.5), which verifies the benefits of textual pre-training knowledge.

**Unified framework vs separate models.** To satisfy different-granularity commands, we compare the performances of training a unified OPA model vs. training multiple separate models in Table 3. In the *3 Single-grained Models* setting (line 6), we train three models respectively to deal with three control granularities, i.e. {*operation*}, {*operation, attribute*}, and {*operation, position, attribute*}. The OPA model reaches a remarkably higher score on the Attr-Acc (93% vs 69%) with better SARI, BLEU4, and ROUGE-L. It demonstrates that our unified input design can alleviate the confusion and heterogeneity of multi-grained commands.

**Difficulty level of various commands.** Table 4 (lines 1-7) displays the performances of different commands, indicating their respective difficulty levels. Generally, the *add* operation proves to be more challenging than *delete*, primarily because it requires the constraint of video content. For the *add* operations, the finer the command granularity (lines 1-4), the higher the *controllable* and *fluency* scores. It reveals that when provided with more detailed control signals, the model can generate desired captions more easily.

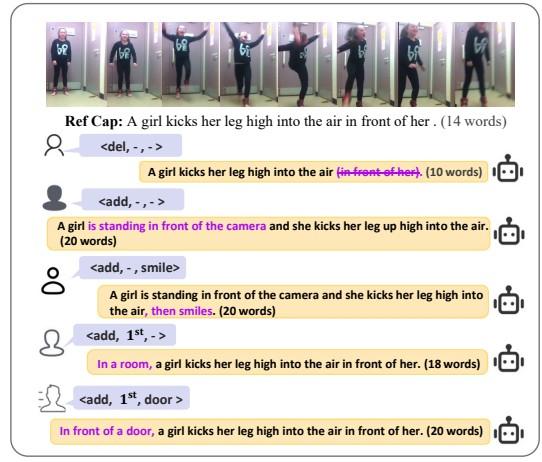

(a) Multi-grained Command Editing

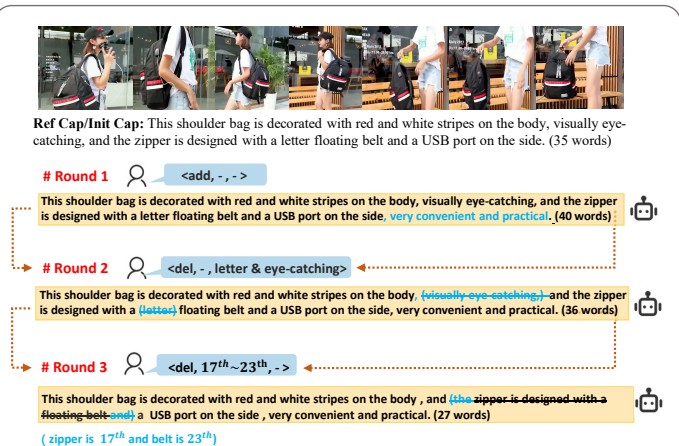

(b) Successive Multi-round Editing

Figure 7: Visualization of multi-grained command editing and successive multi-round editing using the OPA model.

Table 5: The effects of visual modality on the VATEX-EDIT.

| Command | Video | EMScore | SARI | BLEU4 |
|---|---|---|---|---|
| Overall | ✗ | 28.1 | 53.5 | 51.7 |
| | ✔ | 28.7 (+0.6) | 53.9 (+0.4) | 52.2 (+0.5) |
| ⟨del,-,-⟩ | ✗ | 27.0 | 39.4 | 12.1 |
| | ✔ | 28.5 (+1.5) | 40.4 (+1.0) | 13.5 (+1.4) |
| ⟨add,-,-⟩ | ✗ | 28.3 | 41.9 | 10.6 |
| | ✔ | 29.0 (+0.7) | 42.0 (+0.1) | 11.1 (+0.5) |

**Effects of vision modality.** We compare the model performance with and without video input in Table 5. Adding vision modality brings overall metric improvements since it provides visual semantics to guide edited video description generation. For ⟨del,-,-⟩ command, we especially construct challenging samples in which reference captions have misalignments with videos (Section 4.1). With the visual semantics, our model prioritizes removing the video-misalignment contents and achieving a higher EMScore (from 27.0 to 28.5). Similarly, ⟨add,-,-⟩ command requires enriching the original caption referring to the video content.

## 6.4 Quantitative Results

**Multi-grained editing controls.** Provided with various commands, the OPA model can output different edited descriptions to satisfy multi-grained user requests. As Figure 7 (a) shows, our OPA model successfully generates different desired descriptions given the same video, the same reference caption but different commands from coarse-grained (e.g. ⟨add, -, -⟩) to fine-grained (e.g. ⟨add, $1^{st}$, door⟩).

**Successive editing controls.** The OPA model also supports interactive editing with successive controls in the VCE task, depicted in Figure 7 (b). The edited description can serve as the reference caption in the next round to make further editing to satisfy dynamic user demands.

**Human Evaluation.** We further adopt human evaluation to assess the quality of edited video descriptions. We recruit 20 evaluators to score the generated descriptions. We randomly sample 200 test cases

Table 6: Mean score (rated 1-5) of the human evaluation on the two datasets. *Trans.* is short for Pure Transformer.

| Dataset | Model | Control. | Fluency | Vision Align |
|---|---|---|---|---|
| EMMAD-EDIT | Trans. | 2.94 | 3.02 | 3.27 |
| | OPA | 3.98 | 3.85 | 3.76 |
| | GT | 4.67 | 4.37 | 4.22 |
| VATEX-EDIT | Trans. | 4.18 | 4.23 | 3.76 |
| | OPA | 4.36 | 4.43 | 3.93 |
| | GT | 4.48 | 4.34 | 4.41 |

from VATEX-EDIT and 350 cases from EMMAD-EDIT respectively. During the evaluation, we randomly order the edited captions generated from *Pure Transformer baseline*, *OPA*, and *groundtruths (GT)*. The evaluators are asked to rate each description from three aspects on a scale of 1 to 5 points. Table 6 shows the OPA model exceeds the controllable Transformer baseline in three aspects, especially the *controllability*.

## 7 CONCLUSION

We propose a novel multi-modal task named Video Caption Editing (VCE), which aims to automatically edit video descriptions under the guidance of multi-grained user commands. To satisfy diverse and varied user demands, we design the user control signal as a {*operation, position, attribute*} triplet to flexibly cover both coarse-grained and fine-grained controls. We collect two datasets named VATEX-EDIT and EMMAD-EDIT from different domains. We further employ comprehensive metrics to assess fluency, controllability, and vision-text alignment. Finally, we introduce a small specialized model called OPA, an ImgLLM pipeline, and an end-to-end VidLLM to dive into the task challenges and provide good starting points.

**Limitations and Future Work.** This paper primarily introduces appropriate baseline solutions for the VCE task, aiming to provide a thorough analysis. Nonetheless, it falls short of designing architectural innovations, leaving ample room for exploration in the future. Further insights into significant future directions are discussed in Appendix F.

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
