# OpenReview forum: "Edit As You Wish: Video Caption Editing with Multi-grained User Control"
_acmmm.org/ACMMM/2024/Conference — MM2024 Poster_

### Official Review · Reviewer_u3zY · 2024-05-24

**Rating:** 5
**Confidence:** 3

**Summary:**

This paper proposes a controllable user command for the task of video caption editing. Two datasets are established for general and commercial use. A baseline and evaluation metrics of VCE are also developed for evaluation.

**Strengths:**

1. The task VCE and the corresponding datasets established in this paper give a new baseline for more controllable video editing.
2. The evaluation metrics (a combination of fluency, controllability, and text-video alignment) are more sufficient and comprehensive than traditional solutions.
3. Both qualitative and numerical results of this paper show good performance of the baseline.

**Limitations:**

Human annotation of the dataset is costly.  Is it possible to reduce the costs with the help of image editing datasets?

**Suitability:**

3

---

### Official Review · Reviewer_rb12 · 2024-05-25

**Rating:** 5
**Confidence:** 2

**Summary:**

The paper introduces the Video Caption Editing (VCE) task, which aims to enhance video narration by allowing multi-grained user requests for editing captions. It addresses the limitations of existing methods that cannot meet diverse user needs or allow for further edits. The VCE task uses a triplet {operation, position, attribute} to guide revisions. Two datasets, VATEX-EDIT and EMMAD-EDIT, support this task. A specialized model, OPA, is proposed and evaluated against generalist models, revealing challenges in fine-grained multi-modal semantics.

**Strengths:**

- The proposed VCE is interesting and meaningful.
- The paper is well-written, easy to follow, and technically solid.
- The code and instructions are provided with detailed descriptions.

**Limitations:**

- For a more comprehensive comparison, the authors may consider including some large multi-modal language models, either close-souced (e.g., GPT4o, GPT4V) or open-souced (e.g., LLaMA 3, InternVL). To some extent, they will serve as some reference for the "upper bound" of how far the designed VCE model can achieve.
- It would be better to discuss the social impact of the VCE task, both positive and negative.

**Suitability:**

3

---

### Official Review · Reviewer_EESW · 2024-05-26

**Rating:** 5
**Confidence:** 2

**Summary:**

While existing controllable video captioning task can address the various demands of different users, the authors identify several potential shortcomings. This paper introduces a novel task called Video Caption Editing (VCE), which aims to automatically revise existing video descriptions based on multi-grained user requests. To facilitate this new task, the authors have collected two multi-modal datasets named VATEX-EDIT and EMMAD-EDIT. They employ comprehensive metrics to assess fluency, controllability, and vision-text alignment. Additionally, they introduce a small specialized model called OPA, an ImgLLM pipeline, and an end-to-end VidLLM to explore the challenges of the task and provide strong starting points.

**Strengths:**

The task proposed by the authors is highly relevant to practical needs, as it not only considers the use of natural language by users but also takes into account modifications made using a handwriting pad. The introduction of the input triplet framework is particularly intriguing. Furthermore, after collecting and processing two datasets, the authors conducted a thorough and comprehensive analysis using both simple auto-encoder methods and LMM methods, thereby establishing a solid baseline for this task.

The paper is well-written and effectively communicates its ideas. The introduction offers a clear problem statement and task, and the logical flow of the paper ensures that readers can easily follow along.

**Limitations:**

Although the proposed task primarily describes users directly editing video descriptions through multiple rounds of modifications, the dataset mainly uses triplet elements as input. It would be helpful if the authors could provide more detailed explanations of how users utilize natural language for editing (even though the authors vaguely mentioned the use of large language models to convert natural language into triplets in the supplementary materials).

line681 "beam"->"batch"

**Suitability:**

3

---

### Official Review · Reviewer_wjGy · 2024-05-27

**Rating:** 2
**Confidence:** 4

**Summary:**

1. The attempt to extend controllable video description to a multi-grained setting VCE  is promising and valuable.
2. overall paper is clearly presented

**Strengths:**

1. The attempt to extend controllable video description to a multi-grained setting VCE  is promising and valuable.
2. overall paper is clearly presented

**Limitations:**

weakness 1:not well motivated. In the introduction, the authors describe the purpose of VCE (Video Content Enhancement) as assisting merchants in refining product descriptions and conserving effort when dealing with a large number of products. However, I believe there are two issues with this:
1. VCE requires users to input instructions themselves, and when the instructions are related to the video content, users still need to watch the video themselves, which limits the possibility of large-scale automated production.
2. Looking at the current performance of the OPA (Object Proposal and Attention) model (Figure 7), I do not see the necessity for automated production. The effort required for users to input instructions is almost equal to the effort required to edit the content themselves.
weakness2:
Limited preformance. When it comes to large-scale automated production, the model needs to handle many uncommon videos. The performance of the model proposed in this paper, OPA (Object Proposal and Attention), is quite limited on the EMMAD-EDIT dataset compared to methods based on larger models.

**Suitability:**

3

---

### Meta-Review · Area_Chair_n8v7 · 2024-07-01

**Recommendation:** Accept (Poster)
**Confidence:** 5

**Metareview:**

In summary, the reviewers highlight several strengths of the paper, including the introduction of a novel and interesting task, the clarity and logical flow of the writing, and the comprehensive analysis and strong baseline established through datasets and evaluation metrics. Minor concerns are raised regarding the need for more detailed explanations on user interactions, the inclusion of comparisons with larger multi-modal models, the discussion of social impacts, and the cost of human annotation for the dataset.

Overall, most of the ratings are Weak Accept, with one reviewer leaning towards rejection due to concerns about the motivation of the work. Most reviewers appreciate the authors' responses. Considering this, I recommend accepting it for publication.